# Searching for Metabolic Markers of Stroke in Human Plasma via NMR Analysis

**DOI:** 10.3390/ijms242216173

**Published:** 2023-11-10

**Authors:** Nádia Oliveira, Adriana Sousa, Ana Paula Amaral, Gonçalo Graça, Ignacio Verde

**Affiliations:** 1Health Sciences Research Centre (CICS-UBI), University of Beira Interior (UBI), Av. Infante D. Henrique, 6200-506 Covilha, Portugal; nadiaoliveira@fcsaude.ubi.pt (N.O.); amssousa@fcsaude.ubi.pt (A.S.); anapaula@fcsaude.ubi.pt (A.P.A.); 2Section of Bioinformatics, Division of Systems Medicine, Department of Metabolism, Digestion and Reproduction, Imperial College London, South Kensington Campus, London SW7 2AZ, UK

**Keywords:** stroke risk, biomarkers, NMR-based metabolomics, aging

## Abstract

More than 12 million people around the world suffer a stroke every year, one every 3 s. Stroke has a variety of causes and is often the result of a complex interaction of risk factors related to age, genetics, gender, lifestyle, and some cardiovascular and metabolic diseases. Despite this evidence, it is not possible to prevent the onset of stroke. The use of innovative methods for metabolite analysis has been explored in the last years to detect new stroke biomarkers. We use NMR spectroscopy to identify small molecule variations between different stages of stroke risk. The Framingham Stroke Risk Score was used in people over 63 years of age living in long-term care facilities (LTCF) to calculate the probability of suffering a stroke. Using this parameter, three study groups were formed: low stroke risk (LSR, control), moderate stroke risk (MSR) and high stroke risk (HSR). Univariate statistical analysis showed seven metabolites with increasing plasma levels across different stroke risk groups, from LSR to HSR: isoleucine, asparagine, formate, creatinine, dimethylsulfone and two unidentified molecules, which we termed “unknown-1” and “unknown-3”. These metabolic markers can be used for early detection and to detect increasing stages of stroke risk more efficiently.

## 1. Introduction

The latest information shows that 12.2 million people worldwide suffer a new stroke every year and that 66% of all stroke deaths occur in people over 70 years of age [1]. Stroke remains the second leading cause of death and the third cause of death and disability combined [2,3,4]. Ischaemic stroke, caused by a blockage of the bloodstream that leads to brain tissue damage and cell death, accounts for approximately 80% of all strokes [1]. Ischemia can be caused either by embolism, by the formation of atherothrombotic plaques in large vessels, or by decreased blood flow to the brain caused by heart disease (heart failure, myocardial infarction, arrhythmia, atrial fibrillation) [5]. On the other hand, haemorrhagic stroke, which occurs when a vessel ruptures and bleeds into the brain, comprises around 20% and is more lethal than ischaemic stroke [6]. The high rates of mortality and morbidity due to stroke appear to be due to the asymptomatic progression of the disease since, until now, to prevent any type of stroke, clinicians can only try to control the risk factors in the population and can use imaging techniques [7,8]. Stroke’s known risk factors are related to lifestyle, such as unhealthy diet, alcohol abuse, smoking, physical inactivity, and stress [9]. Among the risk factors, the progression of age plays a central role [8], and hypertension and other cardiovascular diseases, such as atrial fibrillation, heart failure, angina pectoris and metabolic diseases (diabetes, dyslipidaemia, obesity), strongly increase the risk of stroke [2,4]

The pathogenesis of stroke has not yet been clearly determined, probably due to its complex aetiology. The discovery of new biomarkers, some of which may be new risk factors, will contribute to a better understanding the pathophysiology of stroke [10]. Since traditional stroke risk factors have been considered insufficient and unable to efficiently predict the occurrence of a stroke, these new biomarkers/risk factors could be very useful to improve diagnosis and to prevent and characterize different stages of progression of the disease [11]. Thus, innovative methodologies to analyse small molecules circulating in the body currently constitute promising tools to improve the diagnostic process and prognosis of the disease and may also provide more information that sheds light on the pathological mechanisms involved [12,13,14]. The identification of new biomarkers may also contribute to the development of new therapeutic approaches and to improve the treatment [15,16]. In this sense, different metabolomic studies were carried out in patients who suffered a stroke. Thus, metabolomic analysis via nuclear magnetic resonance (NMR) allows determining the concentrations of metabolites in different biofluids, and the simultaneous detection and quantification of these molecules at a low cost and high speed [16,17,18].

Some studies on ischaemic stroke using NMR and LC-MS have detected increased levels of asparagine and formate after the occurrence of stroke [19,20,21,22,23]. Some authors have suggested the involvement of pathways related to oxidative stress, such as the homocysteine metabolism and the formation of reactive oxygen species [24,25].

Several authors have found differences in the levels of some branched-chain amino acids (BCAA) related to the occurrence of stroke through metabolomic studies using chromatographic techniques coupled with mass spectrometry. Most of these authors agree that BCAAs have a key role in the maintenance of bioenergetic homeostasis. Wang and colleagues observed a notable decrease in isoleucine levels in the serum of patients with ischaemic stroke [26]. Similar findings were also observed in an animal model of stroke [27], and BCAA levels also appear to decrease in other cerebrovascular and cardiac pathologies [28,29]. On the contrary, other authors, through a prospective study using ultra-high-performance liquid chromatography coupled to MS, found elevated levels of isoleucine in the serum of patients who were later diagnosed with stroke [13].

Other authors found a relationship between renal failure and stroke, evidenced by altered levels of urea and creatinine in both cases [30,31,32]. Some researchers have detected elevated urea levels in stroke victims in the absence of renal dysfunction [33,34]. The mechanisms related to these associations have not yet been clarified, but they are probably related to changes in carbohydrate metabolism [30,31,32].

We analysed plasma from elderly people with varying risks of stroke, in a representative sample of elderly individuals. For the analysis, we obtained spectra of blood plasma via ^1^H NMR, which allowed the quantification of different metabolites, and considered the treatments and concomitant diseases of the participants. Through univariate statistical analysis, we identified potential metabolic biomarkers capable of differentiating stroke risk levels.

## 2. Results

### 2.1. Personal, Sociodemographic and Clinical Data

Study groups may present significant differences in personal, sociodemographic, and clinical characteristics, and these could have an influence on metabolite levels in the groups. In this sense, and for subsequent cross-analysis, Table 1, Table 2 and Table 3 show a summary of several parameters that may be subsequently considered.

Regarding gender, the proportion of women is higher than men in the MSR and HSR groups. On the other hand, the average age of the elderly in the MSR and HSR groups is significantly higher than that of the LSR group. This is consistent with the reality of the distribution of the group of residents in the LTCFs who participated in the study, which are predominantly women over 84 years of age. In contrast, clinical parameters such as body mass index (BMI), blood pressure, heart rate, cholesterol and triglyceride levels in the blood do not show significant differences between the study groups (Table 1).

Table 2 shows the proportions of morbidities in the groups studied. The number of cardiovascular diseases increases from the LSR to the HSR groups (*p* < 0.001), which was expected because the FSRS, the instrument used to constitute the study groups, includes the existence of multiple cardiovascular pathologies (Table 2). Likewise, the prevalence of hypertension (*p* = 0.048), atrial fibrillation (*p* < 0.001), other arrhythmias (*p* < 0.001), heart failure (*p* = 0.021) and acute myocardial infarction (*p* = 0.004) also increases significantly from the low-risk group (LSR) to the high-risk group (HSR) (Table 2). There were no statistically significant differences between stroke risk groups for other cardiovascular diseases such as angina pectoris, atherosclerosis or peripheral vascular disease (Table 2).

Other comorbidities analysed, such as metabolic, respiratory and central nervous system diseases, did not show statistically significant differences between the different groups.

Table 3 shows the percentages of use of some classes of therapeutic drugs for individuals in the stroke risk groups. As expected, there is a significant correlation between the increased risk of stroke and the increased proportion of use of drugs used to treat cardiovascular diseases, such as angiotensin-converting enzyme inhibitors (ACEi) (*p* = 0.006), alpha and beta antagonists (*p* = 0.019), calcium channel blockers (*p* = 0.020), potassium-sparing diuretics (*p* = 0.029) and loop diuretics (*p* = 0.001) (Table 3). On the other hand, there were no significant differences among stroke risk groups in the proportion of use of antiarrhythmics, antianginals, angiotensin receptor antagonists, thiazide diuretics and venotropics (Table 3). As expected, we observed a significant increase in the use of anticoagulants between LSR and HSR groups (*p* = 0.006), as anticoagulants are often used as preventatives when the existence of stroke risk factors is confirmed in patients (Table 3)

In general, the proportion of drugs used to treat metabolic disorders also increases from the LSR to HSR group. However, we only observed a significant increase in the rate of sulfonylurea use (*p* = 0.049). These results are consistent with those documented in Table 1 and Table 2, which show an increase in serum glucose levels and the percentage of elderly people with diabetes between the control group and the HSR group, although this increase was not significant.

Regarding the rate of use of drugs for the treatment of diseases of the central nervous system, acetylcholinesterase and monoamine oxidase inhibitors, N-methyl-D-aspartate antagonists and antipsychotics did not show statistically significant differences between risk groups of stroke. Regarding antiepileptic drugs, we observed a significant decrease in the proportion of use (*p* = 0.022) of LSR to HSR (Table 3). However, we observed an increase in the rate of use of individuals treated with antidepressants from the LSR group to the HSR, despite there being a decrease in the prevalence of depression in HSR (Table 2), although it would be expected that these drugs must also decrease with the increase in stroke risk. However, we observed a significant increase (*p* = 0.028) in the use of these drugs between the LSR and HSR groups (Table 3).

Regarding the use of bronchodilators, we found no significant variation in the proportion of their use with the increased risk of stroke (Table 3).

### 2.2. NMR Spectra Peaks Integration and Identification

Using the NOESY, CPMG, J-Res, HMBC and HSQC pulse sequences, 31 metabolites/signals were identified (Appendix A). A custom R code allowed the integration of isolated peaks of different metabolites from CPMG spectra (Figure 1).

### 2.3. Univariate Metabolomic Analysis of Metabolites Levels

Potential metabolic biomarkers of stroke risk were analysed by comparing the relative concentrations of metabolites identified in plasma NMR spectra of elderly people in stroke risk groups, using univariate statistical analysis (Appendix A).

We observed significant differences between the stroke risk groups in the concentrations of seven metabolites, namely, isoleucine (Ile), asparagine (Asn), creatinine (Cre), dimethyl sulfone (Dms), unknown-1 (Ukn1), unknown-3 (Ukn3) and formate (Figure 2).

Ile levels showed significant changes with increasing risk of stroke (*p* = 0.040). The main differences are related to the reduction in these levels between the LSR and MSR groups (Figure 2A). The increase in Ile levels from LSR to HSR did not reach statistical significance (*p* = 0.962).

We also found a significant increase in Asn (*p* = 0.001), Cre (*p* = 0.016), Dms (*p* = 0.009) and Ukn1 (*p* = 0.024) levels with increasing stroke risk. Differences were found between the LSR and MSR groups and the MSR and HSR groups (Figure 2B–E,G). In the levels of Ukn3 and formate, significant differences were found between the LRS and HSR groups (*p* = 0.036 and 0.031, respectively), but the increase in these metabolites from LSR to MSR did not reach statistical significance (Ukn3 *p* = 0.841; formate *p* = 0.092).

The discriminatory power of stroke risk for the seven metabolites that showed significant differences between the study groups was estimated from the analysis of the area under the receiver operating characteristic curve (AUROC; sensitivity/specificity), using the relative concentrations obtained in univariate analysis. Asn (*p* = 0.002) presented an AUROC above 0.7. Cre (*p* = 0.013), while Dms (*p* = 0.026), Unk1 (*p* = 0.679) and Unk3 (*p* = 0.679) had an AUROC between 0.65 and 0.70. Ile (*p* = 0.140) and formate (*p* = 0.064) had an AUROC below 0.65 and a non-significant *p* value, indicating lower discriminatory power for stroke risk (Figure 3). Thus, these data indicate good performance with good AUROC parameters for Asn, Cre, Dms, Ukn1 and Ukn3 (Figure 3). The information provided by other authors, regarding several of these metabolites as possible biomarkers of stroke (isoleucine, asparagine, creatinine, and formate), has been included in Appendix A.

### 2.4. Comorbidities and Drugs Effect on Plasma Metabolome

Additional statistical evaluations were performed to analyse the possible influence of some diseases and some pharmacological therapies on previously reported differences in metabolite levels between stroke risk groups. We focused this analysis on diseases and therapies that had a different prevalence and rate of use, respectively, between stroke risk groups (Table 2 and Table 3).

As the mean age increases significantly among stroke risk groups (Table 1), we performed a correlation analysis between age and plasma levels of Ile, Asn, Cre, Dms, Ukn1, Ukn3 and formate in all studied individuals and in persons of each stroke risk groups. No correlation was found between the levels of said metabolites and age. Thus, age does not appear to act as a confounding variable in the differences found between stroke risk groups in relation to plasma levels of metabolites.

Furthermore, Table 2 had shown a significant positive correlation between individuals diagnosed with hypertension and the increased risk of stroke by groups. We compared the mean values of Ile, Asn, Cre, Dms, Ukn1, Ukn3 and formate between individuals with and without hypertension in all individuals and in individuals from each of the stroke risk groups. We observed that the mean levels of Asn and Ukn1 were significantly higher in individuals with hypertension (Table 4). Analysing each of the study groups, Asn had a weak but significant correlation with hypertension in the MSR group (r = 0.446). A weak correlation between hypertension and plasma Ukn1 levels was also observed in the LSR group (Table 4). Thus, hypertension is a disease that appears to positively influence the increase in Asn and Ukn1 associated with increased stroke risk (FSRS).

Another type of disease that positively and significantly correlates with the increased risk of stroke by group is arrhythmias (Table 2). We compared the mean values of Ile, Asn, Cre, Dms, Ukn1, Ukn3 and formate between individuals with and without arrhythmia in all individuals and in individuals in each of the stroke risk groups. We observed that there are significantly higher levels of Asn and Ukn1 in individuals with arrhythmia (Table 4) in the cohort, but no significant differences were found in each stroke risk group. We also found that mean Asn levels are significantly higher in people in the atrial fibrillation cohort (Table 4), but no differences were found for Ukn1 or the other metabolites. Examining each of the study groups, Asn shows a weak but significant positive correlation with atrial fibrillation only in the MSR group (r = 0.223) (Table 4). Thus, arrhythmias are pathologies that appear to positively influence the increase in Asn and Ukn1 associated with increased risk of stroke (FSRS) and, among them, atrial fibrillation positively influences Asn.

Regarding acute myocardial infarction, whose prevalence increases from the LSR group to the HSR group (Table 2), we also compared the mean metabolite values between individuals with and without this disease. Mean Ile levels are significantly higher in subjects with acute myocardial infarction (Table 4). Thus, acute myocardial infarction appears to positively influence the increase in Ile associated with increased risk of stroke (FSRS).

The existence of significant but weak correlations indicates that these diseases have a low influence on metabolite levels. However, the significantly different levels of Asn and Ukn1 in people with hypertension and arrhythmia suggest these metabolites as metabolic markers for these conditions. The same can apply in the case of Asn and atrial fibrillation and also for Ile and acute myocardial infarction. No differences were found for any of the other metabolites.

Some pharmacological treatments for some diseases show a positive correlation with an increased risk of stroke (Table 3). In this sense, individuals treated with ACE inhibitors and alpha and beta antagonists increase from the LSR to the HSR groups. However, no mean differences were observed in plasma levels of Ile, Asn, Cre, Dms, Ukn1, Ukn3 and formate between people using or not using ACE inhibitors, nor between individuals using or not using alpha and beta antagonists in the study groups.

Calcium channel blockers (CCB) are antihypertensive medications whose use increases significantly from the LSR to the HSR group. We compared the mean values of metabolites between people who take or do not take CCB and observed that the mean levels of Ile and Ukn1 are significantly higher in individuals who use these drugs (Table 4). The analysis carried out on individuals from each of the study groups showed that the increase in Ile levels does not significantly correlate with the use of CCB, and Ukn1 shows a weak but significant correlation with the use of CCB only in the LSR group (r = 0.276) (Table 4). Thus, the use of CCB appears to positively influence the increase in Ile and Ukn1 associated with increased risk of stroke (FSRS). However, no differences were observed in the mean levels of Ile and Unk1 in the MSR and HSR groups between CCB-treated and non-CCB-treated individuals.

Another type of drug, loop diuretics (LoDiu), is used in an increasing proportion from the LSR to HSR groups (Table 3). In this case, we found a significant increase in the levels of Asn, Cre, Ukn1, Ukn3 and formate in the LoDiu-treated cohort individuals (Table 4). Among each stroke risk group, we observed a positive and significant, although weak, correlation between the percentage of LoDiu use and the levels of Cre (r = 0.310) and Ukn1 (r = 0.338) in the LSR group, and in the MSR only for formate (r = 0.280) (Table 4). Thus, the use of LoDiu appears to positively influence the increase in Asn, Cre, Ukn1, Ukn3 and formate associated with increased stroke risk (FSRS). However, no differences were observed in the mean levels of Asn, Cre, formate, Unk1 and Unk3 in the HSR group between LoDiu-treated and untreated individuals.

Furthermore, the use of anticoagulants, drugs used to help prevent blood clots in people at high cardiovascular risk, also increases with the increased risk of stroke by groups (Table 3). We found a significant increase in the levels of Ile, Ukn1, Ukn3 and formate in individuals medicated with these drugs (Table 4). However, analysing the correlation in each stroke risk group, we found only a significant but weak positive correlation between anticoagulants and formate levels (r = 0.230) (Table 4). Thus, the use of anticoagulants appears to positively influence the increase in Ile, Ukn1, Ukn3 and formate associated with increased risk of stroke (FSRS). However, no differences were observed in the mean levels of Ile, formate, Unk1 and Unk3 in the HSR group between individuals treated and not treated with anticoagulants.

Sulfonylureas (SUR) are antidiabetic drugs whose rate of use increases significantly with the increase in the risk of stroke by group (Table 3). We compared the mean metabolite values between individuals using or not using SUR as therapy and observed that mean Ile levels are significantly higher in individuals treated with SUR (Table 4). However, we found no association between SUR intake and Ile levels in either study group (Table 4).

Table 3 also showed that individuals treated with antiepileptics or antidepressants decrease and increase their use, respectively, from the LSR to the HSR group (Table 3). However, no differences were observed in the mean plasma levels of Ile, Asn, Cre, Dms, Ukn1, Ukn3 and formate between people medicated or not with these types of drugs.

Summarizing the effects of drugs as confounding factors, we can state that the existence of weak correlation parameters and the lack of correlation in various stroke risk groups indicate that the previously mentioned drugs have a weak or no influence on the plasma levels of Ile, Asn, Cre, Ukn1, Ukn3 and formate.

## 3. Discussion

There are currently no clearly established blood metabolomic biomarkers of stroke. Existing metabolomic studies in human serum or plasma were carried out in groups of patients or in populations with different characteristics (for example, age, race, etc.). Most of these studies focused on comparing healthy participants and patients who had suffered a stroke. Many of these studies were performed only with patients in clinical settings and not with a representative population. On the other hand, the procedures used, especially in relation to pre-analytical variables related to sample collection and handling, were very diverse. Furthermore, potential confounders, such as drug therapies or concomitant diseases, are often not considered or evaluated as confounders. To overcome some of these limitations, we analysed plasma from elderly people with varying risk of stroke, in a representative sample of elderly individuals. For the analysis, we obtained spectra of blood plasma via ^1^H NMR, which allowed the quantification of different metabolites, and considered the treatments and concomitant diseases of the participants. Through univariate statistical analysis, we identified potential metabolic biomarkers capable of differentiating stroke risk levels. We found that the relative concentrations of Ile, Asn, formate, Cre, Dms, Unk1 and Unk3 increase with increasing stroke risk (FSRS).

Ile is an essential amino acid that must be acquired through a regular diet. It is a branched-chain amino acid (BCAA), which are compounds whose catabolism allows the formation of acetyl-CoA, propionyl-CoA and succinyl-CoA as end products. BCAAs have been involved in several vital processes, such as the metabolism of key neurotransmitters, protein synthesis and energy production [28,35]. Different evidence supports that BCAAs are also essential in the metabolic response to some diseases, and, for example, different studies have suggested Ile as a biomarker of stroke risk. Thus, BCAAs were positively associated with cardiovascular disease and most strongly associated with stroke in a cohort study using LC-MS/MS (approximately one thousand participants aged 67–70 years) [36]. Zhang et al. indicated that increased plasma Ile levels have a causal effect on the risk of cardioembolic stroke but not on the risk of other stroke subtypes [37]. A study using HPLC-MS/MS on plasma collected from 84 elderly individuals (66–75 years) who suffered ischaemic stroke concluded that Ile decreases after stroke, and the authors suggested that Ile may represent a new link between cerebrovascular, cardiovascular and cardioembolic diseases [27]. Furthermore, other authors have stated that blood levels of BCAAs, such as valine and Ile, are decreased in ischaemic stroke patients compared to healthy individuals [28,38]. However, other authors, using similar methodologies, detected high levels of Ile in the serum of 99 patients (53 years old) who were subsequently diagnosed with stroke. These researchers suggested that hypertension, type 2 diabetes and smoking do not interfere with the aforementioned increase in Ile [13]. Our data show a significant decrease in Ile levels from the LSR to MSR groups, and a non-significant increase in these levels from the LSR to HSR groups, which agrees with data obtained in some of the studies cited previously [13,27,28,38]. Taking this variation in levels into account, it is not uncommon for AUROC analysis to suggest poor Ile performance in discriminating high stroke risk. Other authors, using NMR spectroscopy to measure, over almost 5 years, plasma BCAA levels in women (mean baseline age 55 years), concluded that plasma BCAA levels were positively associated with incident cardiovascular disease, especially myocardial infarction [39]. These findings further suggest an important role for BCAAs in metabolic and bioenergetic homeostasis [27]. Compared to these findings, our results also reveal a significant increase in plasma Ile levels in elderly people with acute myocardial infarction. This suggests that higher Ile concentrations may be associated with an increased risk of stroke in elderly people who have already suffered an acute myocardial infarction.

Our data show that plasma Asn levels increase with increasing stroke risk. This non-essential amino acid, which can be synthesized from oxaloacetate or acquired through the diet, is necessary to support cell function and proliferation in different tissues. In humans, increased plasma Asn levels have been associated with the acute stage of ischaemic stroke but not the chronic stage [20]. Furthermore, targeted metabolomics studies using UPLC-MS/MS in 286 patients aged 63 to 65 years revealed increased serum Asn in mild to moderate ischaemic stroke [22]. Our results show an increase in Asn in elderly people from the LSR to HSR group and from the MSR to HSR group, suggesting that Asn increases before the stroke, and not immediately after the vascular event. These results are also supported by the good power values in the AUROC analysis to discriminate high stroke risk. However, due to the lack of experimental evidence, it is premature to analyse the importance of increased Asn in relation to the increased risk of stroke. It will be necessary to carry out new prospective longitudinal studies with a larger number of participants and in patients who have suffered an ischaemic stroke to analyse the role of Asn.

On the other hand, elevated Asn levels have also been observed in elderly people with hypertension, atrial fibrillation, and other forms of cardiac arrhythmia, which are risk factors for stroke. This suggests that elevated Asn levels may constitute a differentiating biomarker for the increased risk of stroke in people suffering from these cardiovascular diseases. Takemoto et al. found high plasma levels of Asn in a study of elderly people with hypertension. These authors analysed the relationship between high Asn levels and hypertension in a study using intrazosteral injections of the amino acid in freely moving rats, and concluded that Asn causes a direct vasopressor effect combined with a decrease in the glomerular filtration rate [25]. To our knowledge, the relationship between high Asn levels and the appearance of cardiac arrhythmias, especially atrial fibrillation, has never been analysed.

Formate plasma levels also increase with increasing risk of stroke. Formate, which is the simplest carboxylic acid, is essential in the metabolism of single-carbon compounds and the synthesis of nucleic acids and serine [40]. It has been estimated that serine catabolism contributes to approximately 50% of formate production in mice, with the fermentative process carried out by intestinal anaerobic bacteria accounting for the remaining 50% [41]. There are human disorders that manifest alterations in the products of the formate metabolism (e.g., uric acid), in which the role of formate has not yet been elucidated [41]. In a multicentre study with plasma from elderly people collected within 72 h after the onset of suspected ischaemic stroke, and using NMR spectroscopy, elevated formate levels were observed when compared to healthy individuals [23]. The authors suggested that this increase was closely related to folic acid deficiency. In fact, formate is also a final product of homocysteine metabolism. Folic acid deficiency can induce oxidation, due to increased levels of homocysteine, which is associated with oxidation and auto-oxidation of homocysteine and reduced activity of antioxidant enzymes such as superoxide dismutase and glutathione peroxidase [23]. Other authors performed ischaemia/reperfusion studies with microglia and oligodendrocyte cell lines and achieved similar conclusions [42]. In addition, folate deficiency has been found to be very common in ageing [43]. High concentrations of formate have therefore been associated with increased free radical generation. Regardless, these mechanisms have also been associated with other central nervous system diseases such as dementia, Alzheimer’s, and depression [24]. This information agrees with data from the AUROC analysis, which indicate a worse performance of formate in discriminating high risk of stroke.

Plasma levels of Cre, a nitrogenous waste metabolic product, also increase with the intensification of the risk of stroke [30]. Cre is the product of creatine catabolism in muscle. Creatinine and urea are metabolites that are linked through the blood urea nitrogen (BUN) to Cre ratio (BUN/Cre ratio), which is typically used to assess kidney function. In a recent investigation, Peng et al. attempted to establish an association of BUN/Cre with incident stroke and its subtypes, in a large prospective cohort study of an elderly population (57 to 65 years) [30]. These authors concluded that the existence of low levels of the BUN/Cre ratio constitutes a risk factor for the incidence of total and ischaemic stroke, regardless of renal function. They also concluded that high BUN levels increase the risk of total stroke and ischaemic stroke, even in patients with normal kidney function [30]. Other studies also associated high BUN levels with stroke and stroke risk independently from kidney function [33,34], but the mechanisms underlying this association have not yet been clarified. Some authors have suggested the existence of a biological pathway that links elevated BUN and stroke through changes in glucose metabolism and the subsequent occurrence of diabetes [30,31,32]. Other authors have suggested that these two conditions result from decreased water volume in the body [44], which is particularly relevant in aged persons, as dehydration is a very common condition in this group [45]. We did not find a significant association between Cre levels and kidney, metabolic or cardiovascular diseases. The analysis of data from individuals taking loop diuretics showed a significant association among these drugs’ intake and an increased risk of stroke. The mechanism of action of loop diuretics may contribute to a decrease in blood volume, which, according to the hypothesis raised by Coull et al., may induce higher levels of urea in the blood, which was associated with an increased risk of stroke. Our data show an increase in Cre with stroke risk, which causes a decrease in the BUN/Cre ratio, increasing the risk of stroke, which agrees with previously published information [30]. Furthermore, these findings are further supported by the AUROC analysis, which shows that Cre has a good performance in discriminating the increase in stroke risk.

In the body, dimethyl sulfone (Dms), which is commonly present in human blood and cerebrospinal fluid, comes from endogenous metabolism of human methanethiol, dietary sources or bacterial metabolism [46]. Some authors have attributed anti-inflammatory effects to Dms due to its ability to modulate the levels of oxidative stress markers, such as malondialdehyde, protein carbonyl and plasma oxidized glutathione [47]. Furthermore, by mediating the activation of different transcription factors, Dms can regulate the balance of ROS and antioxidant enzymes. Thus, in vitro and in vivo studies have suggested that Dms participates in the inflammatory process and oxidative stress at the transcriptional and subcellular levels [48]. Dms has high permeation capacity in cells and tissues and was used as a carrier or co-transporter for other therapeutic agents. Supplements and foods enriched with Dms are very well tolerated, with almost non-existing adverse effects, and lead to nice results such as relieving joint pain in arthritis, cartilage preservation, decreased muscle damage and improved seasonal allergies, as well as improving skin quality and texture [48]. Some authors have reported that Dms decreases sensitivity to pro-inflammatory stimuli in human cardiac cells, decreasing IL-6 levels. Thus, these authors hypothesised that Dms may protect against cardiac inflammation and may prevent emergency situations due to cardiovascular diseases linked to inflammation [49]. In addition, they have shown that a diet supplemented with Dms induces higher levels of HDL-cholesterol of individuals around 40 years old and enhances the cardiometabolic outcome [50]. However, the effects of Dms on brain and cardiovascular health and disease risk profiles in humans still remain unknown. Our data showed no significant correlation between Dms levels and isolated cardiovascular or metabolic disorders, nor with the intake of any medications. These findings are supported by the AUROC analysis, suggesting a good performance of this metabolite to discriminate high stroke risk. However, to our knowledge, this is the first study to point to Dms as a biomarker of stroke risk. Further studies are necessary to clarify the role of Dms as cardioprotective and its role as possible stroke risk biomarker.

In conclusion, the metabolite markers of stroke risk described can be used for early detection of disease risk, early diagnosis and exploration of pathological mechanisms. Low Ile levels seem to be a promising biomarker to distinguish MSR from LSR, as high levels may be associated with higher stroke risk in elderly people with acute myocardial infarction. On the other hand, positive variations in formate may be influenced by folic acid levels, age and other brain diseases, casting doubt on its specificity as a stroke risk biomarker. The increase in plasma Asn levels with the increase in stroke risk may be related to oxidative stress and has shown to be a promising stroke biomarker, capable of discriminating stroke risk in the elderly with hypertension, atrial fibrillation and other cardiac arrhythmias. Elevated plasmatic Cre levels also appear to be a good biomarker for stroke, even in the absence of renal dysfunction. To better define and elucidate the role of blood BUN/Cre as a biomarker of stroke risk, further laboratory and clinical studies are needed. To our knowledge, this is the first study demonstrating elevated Dms levels as a biomarker of stroke risk. To clarify the role of Unk1 and Unk3, which were also enhanced as promising biomarkers, given the power of AUROC analysis, further research is needed, especially for initial identification of these molecules. Further studies are needed to validate and generalize the applicability of these potential metabolites as novel biomarkers for stroke risk management. 

## 4. Materials and Methods

### 4.1. Study Groups’ Constitution

The study groups were made up of individuals over 64 years old living in LTCFs from an area of ~1000 square kilometres corresponding to three municipalities in Beira Interior (Covilhã, Portugal), which we called EBIcohort.

All procedures realized were previously reviewed and approved by the Ethics Committee of the University of Beira Interior in accordance with the Helsinki convention (Ref. Number CE-UBI-Pj-2017-012). Written informed consent was obtained from cohort participants or their legal representatives. Information regarding the age, sex, clinical and therapeutic data of each participant was provided by the clinical staff of the collaborating LTCF or by community health services. Body mass index (BMI; kg/m^2^) was calculated using the anthropometric data.

Among the EBIcohort participants, all individuals who presented one of the following conditions were excluded: (1) cognitive impairment due to trauma, infection or other causes of brain dysfunction; (2) diagnosis of psychiatric disorders that may contribute to changes in brain function; (3) treatment with aggressive pharmacotherapy with antipsychotics, anticonvulsants, antiretroviral therapy or antiemetics; (4) existence of severe haematological disorders with severe abnormalities in the proportion of white blood cells or severe anaemia; (5) people who have already suffered some type of stroke.

The study groups were formed based on the Framingham Stroke Risk Score (FSRS) values. To calculate each participant’s FSRS, clinical information was used, and a 5-year time window was considered to estimate the probability of suffering a stroke [51,52]. According to the FSRS readjustment carried out in 2017, modifiable and non-modifiable risk factors inherent to stroke were used, such as age, sex and systolic blood pressure; use of antihypertensives; presence or absence of cardiovascular diseases; diabetes; and history of tobacco use [51]. Individuals were divided into 3 groups, taking in account the following risk categories: low stroke risk (<5%; LSR or control), moderate stroke risk (5–20%; MSR) and high stroke risk (>20%; HSR).

Serum samples were used to measure some biochemical parameters classically used in health settings using kits optimized for this purpose (BioSystems Human, Palo Alto, CA, USA): cholesterol oxidase/peroxidase, direct detergent HDL-cholesterol and glycerol phosphate oxidase/peroxidase. LDL values were calculated using the Friedewald formula.

### 4.2. Sample Preparation

Venous blood samples from overnight-fasted individuals were collected in heparin tubes, which were centrifuged for 10 min at room temperature and 1500× *g*. Still within 30 min after blood collection, four plasma aliquots from each participant (400 µL) were stored at −20 °C and, in less than 3 h, were cryopreserved (−80 °C freezer).

To perform the NMR analysis, 300 µL thawed plasma samples were mixed with 600 µL of saline solution (0.9% NaCl in 10% D_2_O); samples were centrifuged for 5 min at 25 °C and 4500× *g*, and 600 µL of the resultant supernatant was set into NMR tubes (5 mm).

### 4.3. NMR Data Acquisition and Processing

Plasma NMR spectra were acquired at 310 K on an AVANCE III 600 MHz NMR spectrometer equipped with a quadruple resonance cryoprobe with an automated sample changer (Bruker SampleJet). For each plasma sample, two 1D pulse sequences were acquired. For Carr–Purcell–Meiboom–Gill (CPMG) pulse sequence (cpmgpr1d; Bruker BioSpin), we used 32 scans, 73.728 data points, spectral width of 20.0286 ppm and relaxation delay of 4 s. Each free induction decay was zero-filled to 64 k points and multiplied by a 0.3 Hz exponential line-broadening function before Fourier transformation. We also acquired Overhauser effect spectrum with pre-saturation (1D NOESY-presat; noesygppr1d, Bruker library) pulse sequence, using 32 scans, 98.304 data points, spectral width of 30.0429 ppm, acquisition time of 2.7 s, relaxation delay of 4 s and mixing time of 100 ms and with water peak suppression.

The NMR spectra were automatically phased, baseline corrected and the chemical shifts were internally referenced to the alpha-glucose anomeric doublet at 5.23 ppm using TopSpin 3.1 software (Bruker, Rheinstetten, Germany). Two-dimensional (2D) ^1^H-^1^H J-resolved (J-res) spectra were also acquired to improve the identification of metabolites. TopSpin 3.1 software was used to control the spectrometer and for data pre-processing. (Bruker Biospin).

Two-dimensional (2D) NMR spectra such ^1^H-^13^C Heteronuclear Single-Quantum Correlation Spectroscopy (HSQC) and Heteronuclear Multiple Bond Correlation (HMBC) were acquired for the representative pool sample to help in the identification of metabolites present in plasma samples. Appendix A shows all overlapping spectra for stroke groups to better demonstrate the spin systems that were integrated and the way in which they were integrated (Appendix A).

### 4.4. Statistical Analysis

NMR data were managed using R statistical software (Version 4.2.2, R Foundation for Statistical Computing, Vienna, Austria) and processed through AlpsNMR R package (version 4.0.4). To avoid the introduction of potential confounder factors in statistical analysis, we excluded the water region between 4.4 ppm and 5.2 ppm. To perform univariate analysis, we have used CPMG spectra. We detected and normalized spectral peaks using Probabilistic Quotient Normalization (PQN). Spectral peak assignment and metabolite identification were carried out by matching chemical shift and peak multiplicity with information from the literature and the Human Metabolome Database (HMDB) (Appendix A).

After assignment, NMR peaks were integrated separately into metabolite signals with good definition. From the data on the different metabolites obtained, some outliers were detected and excluded. The resulting relative concentrations were analysed using One-way ANOVA test (OWAT; IBM Corp. released 2021, IBM SPSS Statistics for Windows, Version 28.0. Armonk, NY, USA: IBM Corp) to check the existence of different mean concentration among the different study groups.

To analyse and compare personal, sociodemographic and clinical data such as gender, age, BMI, systolic blood pressure, comorbidities and medication, we apply test such as the OWAT, Kruskal–Wallis test (KWT) or Fisher’s exact test (FET). To examine differences in the metabolite levels and other variables or correlations among different variables, OWAT or Spearman’s rank correlation (SRC) were used. Box–violin–scatter plots representative of the results concerning metabolites were generated. Analysis of the area under the receiver operating characteristic curve (AUROC) was performed to validate the possible biomarkers (IBM Corp. released 2021, IBM SPSS Statistics for Windows, version 28.0. Armonk, NY, USA: IBM Corp).

## Figures and Tables

**Figure 1 ijms-24-16173-f001:**
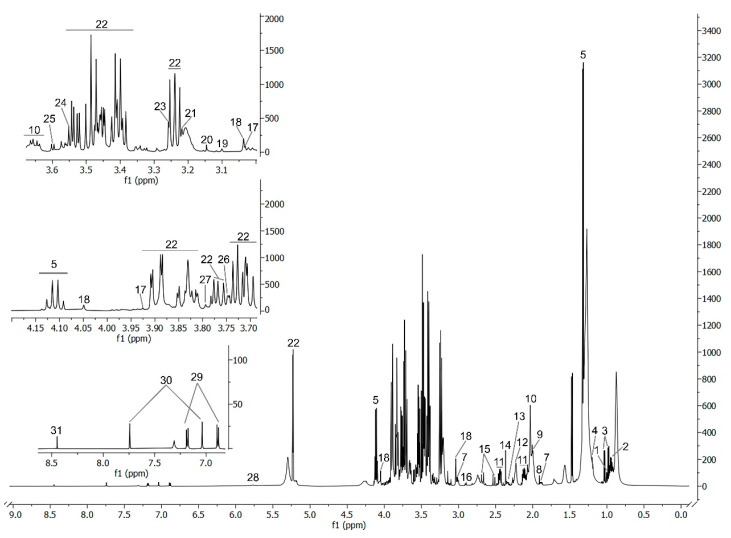
The 600 MHz ^1^H-NMR CPMG spectra from plasma pool sample. Numbers indicate the following metabolites: 1—Isoleucine; 2—Leucine; 3—Valine; 4—3-Hydroxybutyric acid; 5—Lactic Acid; 6—Alanine; 7—Lysine; 8—Acetate; 9—Proline; 10—Glycerol; 11—Glutamine; 12—Acetone; 13—Glutamate; 14—Pyruvate; 15—Citrate; 16—Asparagine; 17—Creatine; 18—Creatinine; 19—Unknown-1; 20—Dimethyl sulfone; 21—Unknown-2; 22—Glucose; 23—Betaine; 24—Glycine; 25—Threonine; 26—Unknown-3; 27—Unknown-4; 28—Urea; 29—Tyrosine; 30—Histidine; 31—Formate.

**Figure 2 ijms-24-16173-f002:**
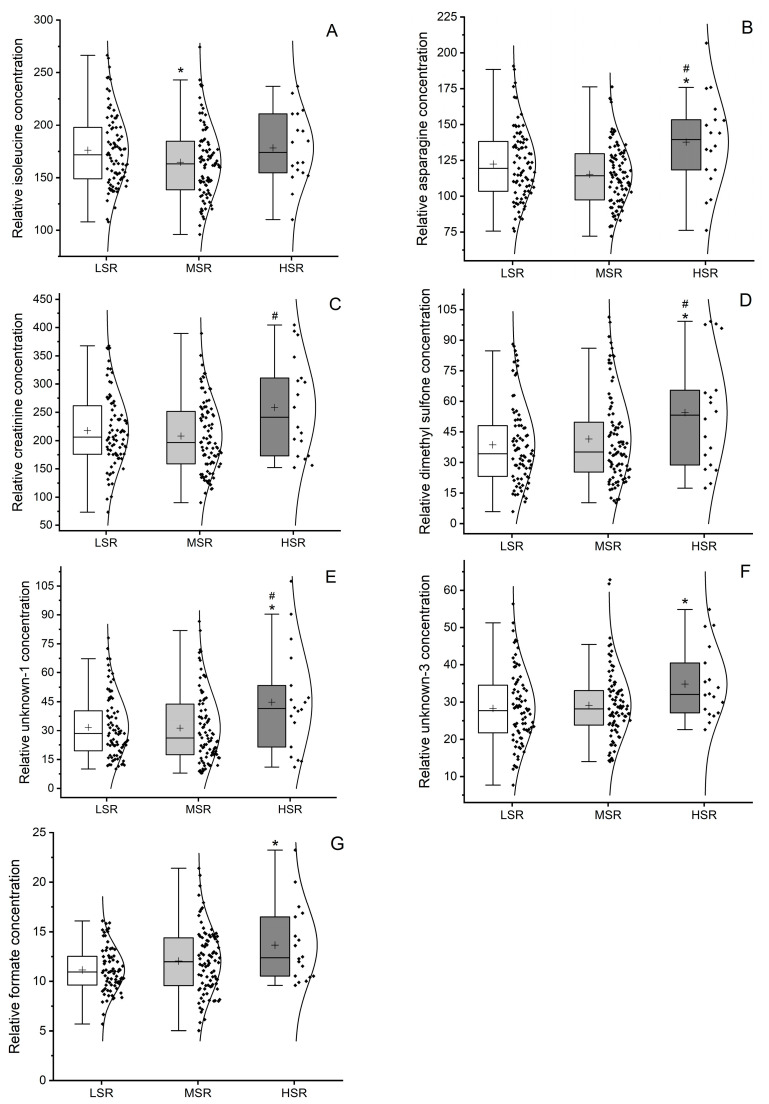
Box–violin–scatter plots representing the variation in relative concentrations of isoleucine (**A**), asparagine (**B**), creatinine (**C**), dimethyl sulfone (**D**), unknown-1 metabolite (**E**), unknown-3 metabolite (**F**) and formate (**G**) among the study groups, namely low stroke risk (LSR, control), moderate stroke risk (MSR) and high stroke risk (HSR). “+” symbols in plots indicate the mean values. Statistical significance was analysed by using One-way ANOVA test. Differences with the control group (LSR) were accessed by using post hoc Dunnet test (* *p* < 0.05), and group-pairs MSR and HSR differences were analysed by using post hoc Tukey test (^#^ *p* < 0.05).

**Figure 3 ijms-24-16173-f003:**
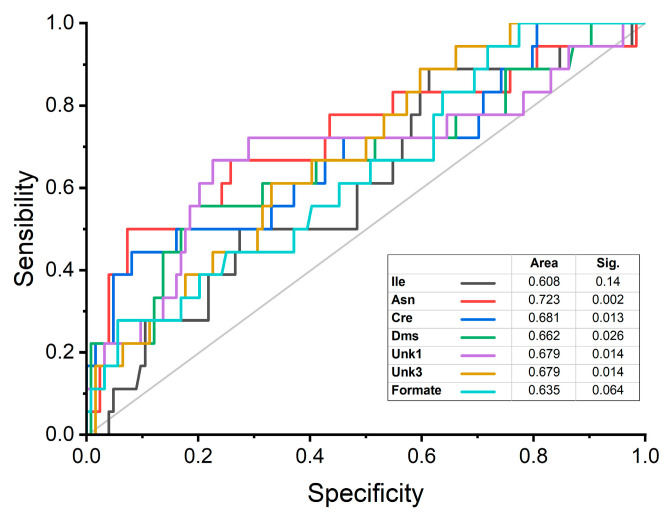
AUROC analysis to discriminate stroke risk for isoleucine (Ile), asparagine (Asn), creatinine (Cre), dimethyl sulfone (Dms), unknown-1 (Unk1), unknown-3 (Unk3) and formate relative concentrations.

**Table 1 ijms-24-16173-t001:** Characterization of personal, sociodemographic and clinical information in the study groups, namely low stroke risk (LSR, control), moderate stroke risk (MSR) and high stroke risk (HSR). Data are expressed as “mean ± s.e.m”, except for gender, expressed as “percentage (*n*)”. Statistical significance was analysed using Kruskal–Wallis test (KWT), One-way ANOVA test (OWAT) and Fisher’s exact test (FET), depending on the homogeneity of variance and the existence of nominal variables, (* *p* < 0.05).

Conditions	LSR	MSR	HSR	*p*-Value	Statistical Test
Number (*n*)	85	94	18		
Age (years)	79.6 ± 0.88	88.2 ± 0.57	90.7 ± 1.28	<0.001 *	KWT
Gender, Female, (%, (*n*))	52.9 (45)	79.8 (75)	77.8 (14)	<0.001 *	FET
Systolic blood pressure (mmHg)	127.6 ± 2.27	122.4 ± 2.04	127.4 ± 4.26	0.199	OWAT
Diastolic blood pressure (mmHg)	67.2 ± 1.32	69.9 ± 1.13	71.8 ± 2.47	0.153	OWAT
Heart rate (bpm)	72.7 ± 1.18	72.7 ± 1.05	71.7 ± 3.22	0.932	OWAT
Body mass index (kg/m^2^)	26.1 ± 0.61	27.1 ± 0.55	28.5 ± 1.32	0.174	OWAT
Serum glucose (mg/dL)	98.3 ± 2.69	99.4 ± 3.13	104.4 ± 8.14	0.740	OWAT
Serum total cholesterol (mg/dL)	165.3 ± 5.06	168.4 ± 4.72	159.2 ± 9.73	0.738	OWAT
Serum HDL-cholesterol (mg/dL)	54.4 ± 1.49	56.3 ± 1.33	59.3 ± 5.01	0.399	OWAT
Serum LDL-cholesterol (mg/dL)	89.6 ± 4.35	89.8 ± 4.34	79.7 ± 7.79	0.651	OWAT
Serum triglyceride (mg/dL)	106.4 ± 4.58	111.6 ± 6.13	100.9 ± 13.66	0.399	OWAT

**Table 2 ijms-24-16173-t002:** Characterization of the main morbidities in stroke risk groups, namely low stroke risk (LSR, control, *n* = 85), moderate stroke risk (MSR, *n* = 94) and high stroke risk (HSR, *n* = 18). COPD, Chronic obstructive pulmonary disease. Data are expressed as “percent (*n*)”. Statistical significance was analysed by using Fisher’s exact test (FET) due to the existence of nominal variables. One-way ANOVA test (OWAT) was used for comparing number of cardiovascular diseases means among groups and data are expressed as “Mean ± s.e.m” (* *p* < 0.05).

Diseases	LSR	MSR	HSR	*p*-Value	Test
**Cardiovascular diseases**					
Hypertension	62.4 (53)	79.8 (75)	83.3 (15)	0.048 *	FET
Atrial fibrillation	2.4 (2)	11.7 (11)	55.6 (10)	<0.001 *	FET
Other Arrythmias	14.1 (12)	19.1 (18)	61.1(11)	<0.001 *	FET
Heart Failure	15.3 (13)	26.6 (25)	(44.4 (8)	0.021 *	FET
Angina pectoris	7.1 (6)	10.6 (10)	(5.6) (1)	0.123	FET
Atherosclerosis	3.5(3)	3.2 (3)	5.6 (1)	0.232	FET
Valvulopathies	1.2 (1)	4.3 (4)	11.1(2)	0.059	FET
Peripheral vascular disease	7.7 (6)	7.8 (7)	5.6 (1)	1.000	FET
Acute myocardial infarction	2 (2.4)	2 (2.1)	4 (22.2)	0.004 *	FET
Cardiovascular diseases (*n*, $\bar{x}$ ± s.e.m)	0.94 ± 0.09	1.32 ± 0.09	2.06 ± 0.24	<0.001 *	OWAT
**Metabolic diseases**					
Diabetes	25.9 (22)	27.7 (26)	33.3(6)	0.361	FET
Dyslipidaemia	38.8 (33)	46.8 (44)	50.0 (9)	0.081	FET
**Respiratory diseases**					
COPD	5.9 (5)	9.6 (9)	5.6 (1)	0.729	FET
Asthma	1.2 (1)	3.2 (3)	5.6 (1)	0.380	FET
**Central nervous system diseases**					
Depression	16.5 (14)	16.0 (15)	5.6 (1)	0.587	FET

**Table 3 ijms-24-16173-t003:** Characterization of the main drugs used to treat comorbidities in the elderly from the study groups, namely low stroke risk (LSR, control, *n* = 85), moderate stroke risk (MSR, *n* = 94) and high stroke risk (HSR, *n* = 18). Data are expressed as “percent (*n*)”. Statistical significance was analysed by using Fisher’s exact test (FET) due to the existence of nominal variables (* *p* < 0.05).

Treatments	LSR	MSR	HSR	*p*-Value	Test
**Treatments for cardiovascular diseases**					
Antiarrhythmics	2.4 (2)	4.3 (4)	0 (0)	0.114	FET
Anti-anginal	9.4 (8)	14.9 (14)	16.7 (3)	0.087	FET
ACEi	9.4 (8)	20.2 (19)	33.3 (6)	0.006 *	FET
Angiotensin receptor antagonists	34.1 (29)	40.4 (38)	55.6 (10)	0.060	FET
Alpha and Beta blockers	18.88 (16)	16.0 (15)	38.9 (7)	0.019 *	FET
Calcium channel blockers	23.5 (20)	14.9 (14)	33.3 (6)	0.020 *	FET
Potassium sparing diuretics	2.4 (2)	7.4 (7)	11.1(2)	0.029 *	FET
Loop diuretics	29.4 (25)	40.4 (38)	72.2 (13)	0.001 *	FET
Thiazide diuretics	20.0 (17)	20.2 (19)	16.7 (3)	0.159	FET
Venotropics	10.6 (9)	12.8 (12)	11.1 (2)	0.152	FET
Anticoagulants	41.2 (35)	53.2 (50)	77.8 (14)	0.006 *	FET
**Treatments for metabolic diseases**					
Sulfonylureas	4.7 (4)	3.2 (3)	11.1 (2)	0.049 *	FET
Biguanides	14.1 (12)	18.1 (17)	11.1 (2)	0.136	FET
DPP-4 inhibitors	16.5 (14)	14.9 (14)	22.2 (4)	0.117	FET
Insulin	5.9 (5)	4.3 (4)	5.6 (1)	0.120	FET
Statins	35.3 (30)	39.4 (37)	50.0 (9)	0.107	FET
**Treatments for respiratory diseases**					
Bronchodilators	10.6 (9)	17.0 (16)	16.4 (3)	0.087	FET
**Treatments for CNS diseases**					
Acetylcholinesterase inhibitors	10.6 (9)	10.6 (10)	5.6 (1)	0.142	FET
Monoamine oxidase inhibitors	0.5 (1)	0 (0)	0 (0)	0.052	FET
NMDA antagonist	7.1 (6)	12.8 (12)	5.6 (1)	0.080	FET
Antiepileptics	12.9 (11)	6.4 (6)	0 (0)	0.022 *	FET
Antipsychotics	32.9 (28)	26.6 (25)	27.8 (5)	0.099	FET
Antidepressants	60.0 (51)	69.1 (65)	88.9 (16)	0.028 *	FET

**Table 4 ijms-24-16173-t004:** Correlations and comparisons between changes in metabolite levels and comorbidities or treatments in stroke risk groups. Comorbidities analysed are acute myocardial infarction, hypertension, atrial fibrillation and other arrhythmias. Pharmacologic drug types studied were calcium channel blockers, loop diuretics, anticoagulants, and sulfonylureas. Isoleucine, Ile; asparagine, Asn; creatinine, Cre; dimethyl sulfone, Dms; unknown-1, Unk1; unknown-3, Unk3. ▲ = increase of metabolite levels when the indicated disease or treatment is present; ns = nonsignificant correlation or difference. Statistical significance of correlations was analysed using Spearman’s rank correlation coefficient (* *p* < 0.05; ** *p* < 0.01). Statistical significance of metabolite values was analysed by Student’s *t*-test (* *p* < 0.05; ** *p* < 0.01; *** *p* < 0.001). ns = not significant.

	**Group**	**Asn**			**Unk1**	
**Comparison (metabolites) with and without hypertension**	LSR	ns			▲ **	
	MSR	▲ ***			ns	
	HSR	ns			ns	
	All	▲ *			▲ ***	
**Comparison (metabolites) with and without arrythmias**	LSR	ns			ns	
	MSR	ns			ns	
	HSR	ns			ns	
	All	▲ *			▲ *	
**Comparison (metabolites) with and without atrial fibrillation**	LSR	ns				
	MSR	▲ *				
	HSR	ns				
	All	▲ *				
	**Group**	**Ile**				
**Comparison (metabolites) with and without acute myocardial infarction**	LSR	ns				
	MSR	ns				
	HSR	ns				
	All	▲ *				
	**Group**	**Ile**			**Unk1**	
**Comparison (metabolites) with and without calcium channel blockers treatment**	LSR	ns			▲ *	
	MSR	ns			ns	
	HSR	ns			ns	
	All	▲ *			▲ *	
	**Group**	**Asn**	**Cre**	**Unk1**	**Unk3**	**Formate**
**Comparison (metabolites) with and without loop diuretics treatment**	LSR	ns	▲ **	▲ **	ns	ns
	MSR	ns	ns	ns	ns	▲ **
	HSR	ns	ns	ns	ns	ns
	All	▲ *	▲ *	▲ ***	▲ *	▲ *
	**Group**	**Ile**		**Unk1**	**Unk3**	**Formate**
**Comparison (metabolites) with and without anticoagulants treatment**	LSR	ns		ns	ns	ns
	MSR	ns		ns	ns	▲ *
	HSR	ns		ns	ns	ns
	All	▲ *		▲ *	▲ *	▲ *
	**Group**	**Ile**				
**Comparison (metabolites) with and without sulfonylureas treatment**	LSR	ns				
	MSR	ns				
	HSR	ns				
	All	▲ ***				

## Data Availability

The data supporting the results of this study are available upon request from the corresponding author, I. Verde. The data are not publicly available due to restrictions under data protection law (Portugal), which contain information that could compromise the privacy of research participants.

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
