# Peer review of "Searching for Metabolic Markers of Stroke in Human Plasma via NMR Analysis"

_ijms, 2023, doi:10.3390/ijms242216173_

Round 1
Reviewer 1 Report
Comments and Suggestions for Authors
Authors present an interesting piece of work via an NMR targeted approach in blood plasma biofluid coupled with statistics, accounting for many cofounding factors .
Overall the work is intersting, however, it needs significant major revisions so as to be potentially considered for publication.
-> The writing is very poor and many phrases should be re-written, because in many cases the reader cannot make any sense of what is written. It seemed to me that the paper was not proofread before submission.
In addition many typos should be corrected, see below very few examples:
Page 3 line 136: “experiences into experiments”
line 146: Were also…
CPOD —> COPD etc..
-> It is striking that authors report urea quantification when it is impossible to quantify urea with these pulse sequences (i.e., water presaturation), since Urea protons highly exchange with water anh their signal intensity is highly affected by the pulse sequence, the pH variability etc. Urea should be removed from any NMR/statistical analysis.
-> Even though I disagree with the use of the CPMG spectra for even the extraction of relative concentrations (due to the CPMG pulse sequence effect to the intensities / linewidhts of signals), there is still macromolecular background that urges the use of deconvolution algorithms or other approaches (e.g., https://doi.org/10.1007/s10858-015-9962-3 or https://doi.org/10.1039/D0SC01421D). So, for the NMR signals integration, authors should overlay the whole number of spectra and they could color them by group so as to demonstrate the spin systems that were integrated and the way they were integrated.
-> Overall, authors put a significant effort for the statistical analyses, however, it would be nice for the reader to explain (and also report) the reason for not doing MVA analyses (?).
Comments on the Quality of English Language
-> The writing is very poor and many phrases should be re-written, because in many cases the reader cannot make any sense of what is written. It seemed to me that the paper was not proofread before submission.
In addition many typos should be corrected, see below very few examples:
Page 3 line 136: “experiences into experiments”
line 146: Were also…
CPOD —> COPD etc..
Author Response
Comments and Suggestions for Authors
Authors present an interesting piece of work via an NMR targeted approach in blood plasma biofluid coupled with statistics, accounting for many cofounding factors.
Overall the work is interesting, however, it needs significant major revisions so as to be potentially considered for publication.
-> The writing is very poor, and many phrases should be re-written, because in many cases the reader cannot make any sense of what is written. It seemed to me that the paper was not proofread before submission.
In addition, many typos should be corrected, see below very few examples:
Page 3 line 136: “experiences into experiments”
line 146: Were also…
CPOD —> COPD etc..
-> It is striking that authors report urea quantification when it is impossible to quantify urea with these pulse sequences (i.e., water presaturation), since Urea protons highly exchange with water and their signal intensity is highly affected by the pulse sequence, the pH variability etc. Urea should be removed from any NMR/statistical analysis.
-> Even though I disagree with the use of the CPMG spectra for even the extraction of relative concentrations (due to the CPMG pulse sequence effect to the intensities / linewidhts of signals) , there is still macromolecular background that urges the use of deconvolution algorithms or other approaches (e.g., https://doi.org/10.1007/s10858-015-9962-3 or https://doi.org/10.1039/D0SC01421D). So, for the NMR signals integration, authors should overlay the whole number of spectra and they could color them by group so as to demonstrate the spin systems that were integrated and the way they were integrated.
-> Overall, authors put a significant effort for the statistical analyses, however, it would be nice for the reader to explain (and also report) the reason for not doing MVA analyses (?) .
ANSWER: The pH variability that most affects the urea signal is not a problem in our study, since in human plasma samples pH variations are insignificant. However, we recognize that the water suppression imposed by the NMR pulse sequences used in our study likely affects the urea signal intensity. Therefore, all NMR and statistical data regarding urea were removed in the new version of the manuscript.
Our study was carried out following standardized operating procedures for the production, conservation, and manipulation of the collected samples, which were recommended to us by the scientific support services of the company selling the equipment and by collaborating researchers. We also meticulously utilize the endorsed NMR pulse sequences for the metabolomics studies performed on human plasma (doi.org/10.1002/anie.201804736; doi.org/10.1038/nprot.2007.376; doi.org/10.1016/j.nbt.2019.04 0.004). As suggested, and to better demonstrate the spin systems that were integrated and the way in which they were integrated, we added an image of all overlapping spectra with different colors by study groups (blue, black, gray).
We made a new figure by superimposing the entire number of spectra and colored them by group. The result constitutes supplementary figure S3.
Concerning univariate and multivariate analysis, initially, we were more interested in univariate analysis, because this allows us to understand the distribution of values for the variables that this type of analysis considers most important, in this case stroke risk and metabolites, which allows us to analyze the relationship between them, sometimes carrying out paired analyzes or analyzing the crosstalk of a variable in a relationship between two other variables. We think this is essential before performing multivariable analysis.
In fact, in we begin the analysis in related areas by performing multivariate analyses, such as multiple linear regression models or principal component analysis, and we feel that, in many cases, "the relationships" are unclear or that could be the result of mathematical constructs. On the other hand, if the multivariate analysis is not carried out very carefully, it may lead to underestimating or giving less value to variables that may be important, and the opposite can also occur. In this context, we first decided to move forward with univariate analysis to better understand the most important variables, which does not exclude us from carrying out multivariate analyzes in the future but with more data related to other types of molecules (lipids or proteins, for example).
Comments on the Quality of English Language
-> The writing is very poor and many phrases should be re-written, because in many cases the reader cannot make any sense of what is written. It seemed to me that the paper was not proofread before submission.
In addition, many typos should be corrected, see below very few examples:
ANSWER: The manuscript was proofread and deeply revised before resubmission. Many parts of the manuscript were rewritten. Errors were corrected.
Page 3 line 136: “experiences into experiments”
line 146: Were also…
CPOD —> COPD etc..
ANSWER: The mentioned errors have been corrected in the current version of the manuscript.
Reviewer 2 Report
Comments and Suggestions for Authors
The manuscript entitled “Seeking for stroke metabolic markers in human plasma by 2 NMR analysis” aimed to analyse plasma from elderly people with variable risk of stroke, in a sample of an elderly population, considering the treatments and concomitant diseases of the participants, using quantitative 1H NMR spectroscopy in conjunction with univariate statistical analysis and identified potential metabolic biomarkers capable of differentiating stroke risk levels. The work that is presented if of great relevance, nevertheless, some suggestions are necessary for its proper publication.
1) The references are not properly cited into the text, they must be placed before the “dot” not after. Please correct throughout the manuscript.
2) Grammar must be reviewed and corrected throughout the manuscript, authors are encouraged to review their manuscript by a native English speaker.
3) A table which presents the different biomarkers that have been reported by several authors related to strokes is advised to be presented to provide a better context to the reader.
4) Paragraph- lines 79-87 could fit better in discussion section, where this information may be properly discussed along with the results in this investigation.
5) In the section Materials and Methods, 2.1. Study groups constitution; inclusion, exclusion and elimination criteria for each study group should be broadly described, please address.
6) In the section Materials and Methods, 2.2. Sample preparation; correct line 136 experiences to experiments.
7) Results are well presented nevertheless authors did not present chemometric analysis, is suggested to present, due to the nature of the study and the quality of the information presented in the manuscript, also in the discussion it may be useful to picture the differences between groups of analysis with a better understanding for the reader and thus a better presentation of the results, that may illustrate differences that are not shown with conventional statistics.
8) The discussion is well stated. A comparative table with other studies might be helpful in order to provide a better overview of the studies that are already reporting this biomarkers for this same disease.
9) Some of the references have the DOI missing, please add.
10) Information about Author Contributions, Funding, Institutional Review Board Statement, Informed Consent Statement, Data Availability Statement, Acknowledgments, Conflicts of Interest is missing. Please add.
Comments on the Quality of English LanguageThe manuscript entitled “Seeking for stroke metabolic markers in human plasma by 2 NMR analysis” aimed to analyse plasma from elderly people with variable risk of stroke, in a sample of an elderly population, considering the treatments and concomitant diseases of the participants, using quantitative 1H NMR spectroscopy in conjunction with univariate statistical analysis and identified potential metabolic biomarkers capable of differentiating stroke risk levels. The work that is presented if of great relevance, nevertheless, some suggestions are necessary for its proper publication.
1) The references are not properly cited into the text, they must be placed before the “dot” not after. Please correct throughout the manuscript.
2) Grammar must be reviewed and corrected throughout the manuscript, authors are encouraged to review their manuscript by a native English speaker.
3) A table which presents the different biomarkers that have been reported by several authors related to strokes is advised to be presented to provide a better context to the reader.
4) Paragraph- lines 79-87 could fit better in discussion section, where this information may be properly discussed along with the results in this investigation.
5) In the section Materials and Methods, 2.1. Study groups constitution; inclusion, exclusion and elimination criteria for each study group should be broadly described, please address.
6) In the section Materials and Methods, 2.2. Sample preparation; correct line 136 experiences to experiments.
7) Results are well presented nevertheless authors did not present chemometric analysis, is suggested to present, due to the nature of the study and the quality of the information presented in the manuscript, also in the discussion it may be useful to picture the differences between groups of analysis with a better understanding for the reader and thus a better presentation of the results, that may illustrate differences that are not shown with conventional statistics.
8) The discussion is well stated. A comparative table with other studies might be helpful in order to provide a better overview of the studies that are already reporting this biomarkers for this same disease.
9) Some of the references have the DOI missing, please add.
10) Information about Author Contributions, Funding, Institutional Review Board Statement, Informed Consent Statement, Data Availability Statement, Acknowledgments, Conflicts of Interest is missing. Please add.
Author Response
Comments and Suggestions for Authors
Comments on the Quality of English Language
The manuscript entitled “Seeking for stroke metabolic markers in human plasma by 2 NMR analysis” aimed to analyse plasma from elderly people with variable risk of stroke, in a sample of an elderly population, considering the treatments and concomitant diseases of the participants, using quantitative 1H NMR spectroscopy in conjunction with univariate statistical analysis and identified potential metabolic biomarkers capable of differentiating stroke risk levels. The work that is presented if of great relevance, nevertheless, some suggestions are necessary for its proper publication.
1) The references are not properly cited into the text, they must be placed before the “dot” not after. Please correct throughout the manuscript.
ANSWER: This writing error was corrected in the new version of the manuscript.
2) Grammar must be reviewed and corrected throughout the manuscript, authors are encouraged to review their manuscript by a native English speaker.
ANSWER: Grammar has been reviewed and corrected throughout the manuscript by a proficient English speaker.
3) A table which presents the different biomarkers that have been reported by several authors related to strokes is advised to be presented to provide a better context to the reader.
ANSWER: As suggested in point 8, a comparative table with the other studies reporting these biomarkers for stroke or stroke risk was included in the new version of the manuscript as supplementary table S3. A more comprehensive table including all biomarkers suggested by the different authors in relation to stroke would be appropriate for a review article, or a meta-analysis article. We think that table S3 already give a better context to the reader and provide a better overview of the studies that are already reporting these biomarkers
4) Paragraph- lines 79-87 could fit better in discussion section, where this information may be properly discussed along with the results in this investigation.
ANSWER: The content of these alignments was transferred to the beginning of the discussion and contextualized. The last part of the introductory section was adapted according to this change.
5) In the section Materials and Methods, 2.1. Study groups constitution; inclusion, exclusion and elimination criteria for each study group should be broadly described, please address.
ANSWER: The "constitution of study groups" is a subsection of "materials and methods" section. The constitution of the groups, based on the FSRS values, is referred to in paragraph 4 of this section. The general exclusion criteria are referred to in paragraph 3, there are no other inclusion or exclusion criteria than those referred to for all groups, except the use of different FSRS values to include participants in one of the study groups.
6) In the section Materials and Methods, 2.2. Sample preparation; correct line 136 experiences to experiments.
ANSWER: The correction of this error was performed in the new version of the manuscript.
7) Results are well presented nevertheless authors did not present chemometric analysis, is suggested to present, due to the nature of the study and the quality of the information presented in the manuscript, also in the discussion it may be useful to picture the differences between groups of analysis with a better understanding for the reader and thus a better presentation of the results, that may illustrate differences that are not shown with conventional statistics.
ANSWER: Chemometrics is a broad discipline that includes mathematical and statistical methods to enable the analysis and representation of data from a MULTIVARIATE perspective of data analysis. We performed a univariate analysis, not a multivariate analysis that would allow us to include this figure related to chemometrics.
8) The discussion is well stated. A comparative table with other studies might be helpful in order to provide a better overview of the studies that are already reporting these biomarkers for this same disease.
ANSWER: A comparative table with the other studies reporting these biomarkers for stroke or stroke risk was included in the new version of the manuscript as supplementary table S3.
9) Some of the references have the DOI missing, please add.
ANSWER: Most of references had Doi missing, these missing Doi were included in the new version of the manuscript.
10) Information about Author Contributions, Funding, Institutional Review Board Statement, Informed Consent Statement, Data Availability Statement, Acknowledgments, Conflicts of Interest is missing. Please add.
ANSWER: The missing information was added in the new version of the manuscript (Funding, Acknowledgments, Conflicts of Interest, Ethics statement and Data Availability Statement).
“Author Contributions” were indicated in the manuscript submission and are in the face menu of submission:
Conceptualization, Nadia Oliveira, Ana Saraiva Amaral and Ignacio Verde; Methodology, Nadia Oliveira, Adriana Sousa, Ana Saraiva Amaral and Gonçalo Graça; Software, Gonçalo Graça and Ignacio Verde; Validation, Adriana Sousa, Gonçalo Graça and Ignacio Verde; Formal analysis, Nadia Oliveira, Adriana Sousa, Ana Saraiva Amaral, Gonçalo Graça and Ignacio Verde; Investigation, Nadia Oliveira, Adriana Sousa, Ana Saraiva Amaral and Ignacio Verde; Data curation, Nadia Oliveira, Adriana Sousa, Ana Saraiva Amaral and Ignacio Verde; Writing – original draft, Nadia Oliveira; Writing – review & editing, Nadia Oliveira, Gonçalo Graça and Ignacio Verde; Visualization, Adriana Sousa and Ana Saraiva Amaral; Supervision, Ignacio Verde; Project administration, Ignacio Verde; Funding acquisition, Ignacio Verde.